

# Assesment of bone healing after surgical management of odontogenic cysts utilizing fractal analysis—a retrospective cross-sectional study

Ayse Tas[1], Elif Celebi[2] and Zeynep Çukurova Yilmaz[3]

[1] Department of Oral and Maxillofacial Radiology, Faculty of Dentistry, Istanbul Medipol University, Istanbul, Turkey

[2] Department of Oral and Maxillofacial Radiology, Faculty of Dentistry, Bahcesehir University, Istanbul, Turkey

[3] Department of Oral and Maxillofacial Surgery, Faculty of Dentistry, Istanbul Medipol University, Istanbul, Turkey

Corresponding author
Ayse Tas, ayse.tas@medipol.edu.tr

## ABSTRACT

**Objectives**. Odontogenic cysts, originating from inflammatory or developmental processes of the tooth germ epithelium, represent the most common intraosseous pathology in the head and neck region. This retrospective study aimed to evaluate bone healing following the surgical removal of odontogenic cysts using fractal analysis.

**Materials and Methods**. Bone changes in 17 patients who underwent cyst enucleation were assessed using fractal dimension and lacunarity measurements on digital panoramic radiographs obtained before and six months after surgery. Statistical analyses included the Shapiro–Wilk test, paired $t$-test, Wilcoxon signed-rank test, repeated measures ANOVA, and the Friedman test.

**Results**. At six months postoperatively, bone structure differences had largely normalized. In the cyst center, there were no statistically significant differences in fractal dimension or lacunarity ($P > 0.05$). At the cyst edge, fractal dimension remained statistically unchanged ($P = 0.446$), while lacunarity significantly decreased from 0.06 to 0.04 ($P = 0.04$). In unaffected control regions, no significant changes were observed ($P > 0.05$). Preoperative fractal dimension values significantly differed between regions ($P < 0.001$), but these differences were no longer significant postoperatively ($P = 0.077$). Lacunarity values showed no significant regional differences at either time point ($P > 0.05$).

**Conclusions**. Fractal analysis offers valuable insights into bone regeneration and may surpass traditional radiographic evaluations. Despite limitations such as small sample size and potential subjectivity in region of interest (ROI) selection, the results support the utility of fractal dimension and lacunarity in tracking bone healing after cyst surgery.

## INTRODUCTION

Odontogenic cysts arise from either inflammatory or developmental pathological processes associated with the epithelium of the tooth germ. These cysts are the most prevalent intraosseous pathology in the head and neck region (*Murugesan, Vadivel & Ramalingam,*

*2024*; *Wongrattanakarn et al., 2023*; *Tekkesin et al., 2012*). Predominantly occurring in males, odontogenic cysts have the highest incidence during the second and third decades of life. Clinically, these cysts often present as painless swellings, which may be accompanied by cortical expansion and positive findings on needle aspiration. Radiographically, they typically appear as unilocular radiolucent lesions with well-defined borders and are frequently located in the posterior mandible. The most common types of odontogenic cysts include radicular cysts, dentigerous cysts, residual cysts, and odontogenic keratocysts (OKCs) (*Tekkesin et al., 2012*; *Vered & Wright, 2022*; *Santosh, 2020*). Among these, the OKC is particularly noteworthy due to its distinctive clinical behavior and potential for recurrence. This has led the World Health Organization (WHO) to reclassify it as a tumor (keratocystic odontogenic tumor) in 2005 (*Vered & Wright, 2022*). However, in 2017, the WHO reverted this classification, once again categorizing the OKC as a cyst (*Murugesan, Vadivel & Ramalingam, 2024*; *Soluk-Tekkeşin & Wright, 2018*).

Treatment approach for odontogenic cysts is enucleation (*Vered & Wright, 2022*). Additionally, decompression or marsupialization treatments are increasingly considered viable alternatives, particularly for large lesions, due to their superior outcomes in preserving critical anatomical structures or developing teeth. The advantages of these procedures include a reduced likelihood of injury to adjacent anatomical tissues, decreased need for hospitalization and extensive surgical interventions, as well as lower treatment costs. Nonetheless, these methods have drawbacks, such as prolonged treatment periods, the necessity for continuous monitoring, patient compliance with the use of specialized apparatus, and the inability to fully examine the entire lesion histopathologically (*Hauer et al., 2020*; *Akay, Kaya & Zeytinoğlu, 2011*).

Radiographic imaging is routinely used in dental health examinations and is also preferred for monitoring the success of applied treatments. One advanced technique, fractal analysis (FA), provides a quantitative assessment of the microarchitectural structure of trabecular bone using radiographic images. This method employs a mathematical and morphological image processing system based on the fractal dimension (FD) of the bone. Panoramic radiographs can be analyzed by selecting a specific area from the relevant region for FA (*Ozturk et al., 2022*; *Gumussoy et al., 2016*). Another image analysis parameter is lacunarity, which is a characteristic that reflects how gaps are distributed and arranged within an image. In image analysis, it offers supplementary information that goes beyond what the FD reveals. While FD assesses the complexity or roughness of an image, lacunarity provides a deeper understanding of the texture by measuring the heterogeneity or variability in gap distribution. A decrease in lacunarity indicates that the gaps are becoming more evenly and uniformly distributed (*Da Silva et al., 2023*).

The aim of this retrospective study is to evaluate bone healing after the surgical treatment of odontogenic cysts using FA. The hypothesis is that the changes in the trabecular structure in the cyst areas post-treatment will not be statistically significant during the treatment and post-treatment periods.

## MATERIAL AND METHODS

### Patient enrolment and treatment approach

From the cohort of consenting patients between January 1, 2018, and December 31, 2021, a randomized selection process was employed. This process resulted in the inclusion of 17 patients who underwent surgical intervention at the Department of Oral and Maxillofacial Surgery, Faculty of Dentistry, Istanbul Medipol University, for the treatment of radicular, dentigerous, residual cysts, and OKC. All cystic lesions included in this study were surgically enucleated and subsequently confirmed through histopathological examination. No cases underwent decompression or received chemical adjunctive treatment.

Inclusion criteria included patients over 18 years of age with histopathologically confirmed odontogenic cysts and available pre- and post-operative panoramic radiographs. Exclusion criteria encompassed the presence of systemic ailments impacting bone metabolism, as well as the usage of medications that might have influenced bone turnover.

The six-month follow-up interval was selected based on institutional protocol, aiming to allow sufficient time for observable bone remodeling while maintaining consistency across patient records.

### Ethical considerations

This retrospective study was approved by the Istanbul Medipol University Faculty of Medicine Clinical Research Ethics Committee (Approval No: E-10840098-202.3.02-2244, 2024). The study was conducted in accordance with the ethical principles of the Declaration of Helsinki. Since archived anonymized panoramic radiographs were used, the requirement for obtaining informed consent was waived by the ethics committee.

### Data collection and radiographic analysis

All panoramic radiographs were acquired using the same digital panoramic unit (Orthopantomograph® OP100) with standardized exposure settings to ensure consistency in image quality for fractal analysis. Upon securing informed consent, preoperative and 6-month postoperative panoramic radiographs were acquired using an panoramic X-ray unit (Orthopantomograph® OP100; Instrumentarium Imaging, Tuusula, Finland). The radiographic technique employed a 2.5 mm Al equivalent total filtration, operating at 70 kVp and 16 mA for a duration of 17.6 s. The resultant radiographic images were subsequently exported in Tagged Image File Format (TIFF) with dimensions of $2{,}884 \times 1{,}504$ pixels (Centricity Universal Viewer; GE HealthCare, Chicago, IL, USA).

### Image analysis
#### Region of interest selection

The analytical process was conducted utilizing the ImageJ version 1.54f computer software (National Institute of Health, Bathesda, MD, USA). A manual selection of a region of interest (ROI) measuring $50 \times 50$ pixels was carried out. These ROIs were meticulously chosen from the center of the cyst, proximate to the inner margin of the lesion, and symmetrically from the unaffected bone region (Fig. 1). Each ROI was saved and cataloged within the ROI manager, ensuring a standardized procedure for all images obtained

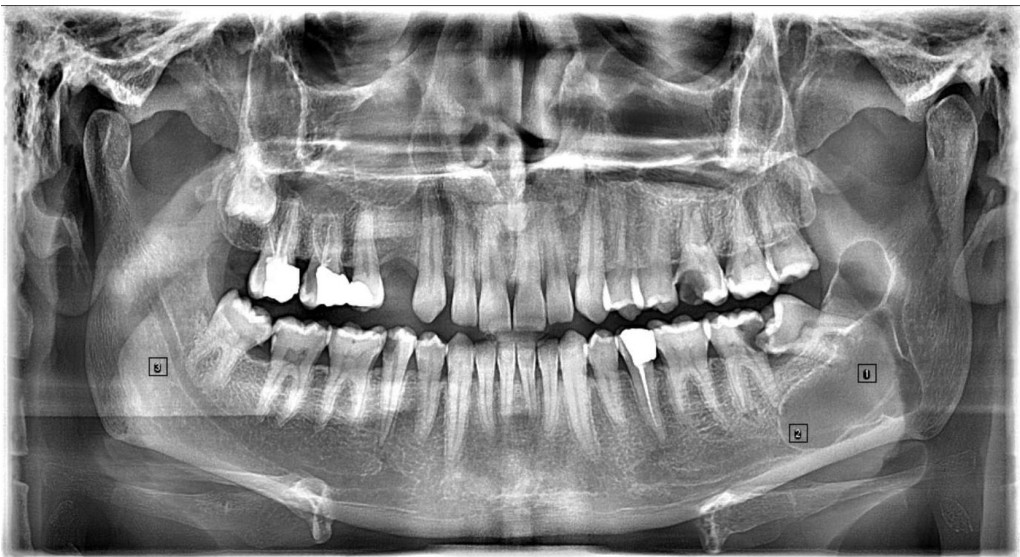

**Figure 1** **Panoramic radiograph showing region of interest (ROI) selections for fractal analysis.** The image demonstrates the location of ROIs selected at the center and edge of the lesion and in the unaffected contralateral region of the mandible. ROIs measured 50 × 50 pixels and were used for calculating fractal dimension and lacunarity.

from a single patient. The selection of ROIs was performed collaboratively by two calibrated operators (A.T. and E.C.), both possessing a background as oral and maxillofacial radiologists with six years of experience.

*Evaluation of fractal dimension and lacunarity values*

The assessment of FD involved a non-binarization approach, employing the FracLac plugin for ImageJ (School of Community Health, Faculty of Science, Charles Sturt University, Australia) to directly compute FD from the grayscale images. The box-counting method was specifically chosen within the plugin, and the "Gray1: Differential" option was designated. For the grid layout, a total of 12 grid positions were defined. All images underwent scanning using this standardized configuration, ultimately yielding FD and lacunarity values (Eizo RadiForce RX240, resolution: 1,200 × 1,600 mp).

Intraclass Correlation Coefficient was used to examine the interobserver agreement of quantitative values before and after treatment. Intraobserver agreement for radiographic parameters was reassessed using eight randomly selected CBCT images over a 4-week interval. The intraclass correlation coefficient (ICC) for lacunarity was 0.849 (95% confidence interval (CI) [0.632–0.943]). Similarly, the ICC for fractal measurements was 0.716 (95% CI [0.373–0.887]), indicating a high degree of agreement ($P < 0.001$).

## Statistical analysis

Data were analyzed in SPSS® Statistics version 23.0 (IBM Corp., Armonk, NY, USA). The suitability of the data for normal distribution was examined with the Shapiro–Wilk test. Dependent samples $t$-test was used to compare quantitative values complying with normal distribution at two times. Wilcoxon test was used to compare quantitative values that did
**Table 1 Comparison of quantitative values between pre-treatment and post-treatment 6th month values in the regions.** This table presents the mean, median, and statistical comparisons of fractal and lacunarity values for center, edge, and unaffected bone regions before and after treatment.

| Region | | Pre-treatment | | Post-treatment | | Test statistics | $p$ |
|---|---|---|---|---|---|---|---|
| | | Mean ± SD | Median (Min–Max) | Mean ± SD | Median (Min–Max) | | |
| Center | Fractal | 1.39 ± 0.1 | 1.4 (1.21–1.58) | 1.37 ± 0.1 | 1.36 (1.18–1.54) | 0.692 | 0.499[*] |
| | Lacunarıty | 0.06 ± 0.04 | 0.04 (0.02–0.17) | 0.06 ± 0.03 | 0.05 (0.02–0.14) | −0.97 | 0.332[**] |
| Edge | Fractal | 1.36 ± 0.11 | 1.34 (1.19–1.56) | 1.38 ± 0.15 | 1.37 (1.1–1.63) | −0.782 | 0.446[*] |
| | Lacunarıty | 0.06 ± 0.03 | 0.06 (0.01–0.12) | 0.04 ± 0.01 | 0.04 (0.02–0.08) | 2.232 | **0.040**[*] |
| Unaffected | Fractal | 1.5 ± 0.1 | 1.5 (1.33–1.66) | 1.45 ± 0.09 | 1.46 (1.27–1.6) | 2.013 | 0.061[*] |
| | Lacunarıty | 0.05 ± 0.03 | 0.04 (0.02–0.15) | 0.04 ± 0.02 | 0.04 (0.02–0.12) | −0.544 | 0.586[**] |

**Notes.**
[*]Dependent samples $t$-test.
[**]Wilcoxon test.
Bold values indicate statistically significant results ($P < 0.05$).

not comply with normal distribution at two times. Repeated Measures Analysis was used to compare normally distributed quantitative values according to regions. Friedman test was used to compare quantitative values that did not show normal distribution according to regions. Analysis results were presented as mean and standard deviation (mean (SD)), and median. The significance level was considered $P < 0.05$.

## RESULTS

This study investigated the effects of treatment on trabecular bone structure using quantitative imaging parameters namely fractal dimention and lacunarity in three different regions of interest.

A total of 17 patients were included in the study: 11 (65%) of these patients were men and six (35%) were women. The average age of the patients with radiographs participating in the study was calculated as 35.6 years. Of the cysts detected on radiographs of these patients, 53% were radicular cysts, 23% were OKC, 12% were residual cysts, 12% were dentigerous cysts (Table 1).

In the center region of cyst, there was no statistically significant difference in quantitative values between pretreatment and six months post-treatment ($P^{\text{fractal}} = 0.499$) ($P^{\text{lacunarity}} = 0.332$) ($P > 0.05$). Similarly, in the edge region, no significant difference was observed in fractal values between pretreatment and six months post-treatment ($P = 0.446$). However, there was a significant difference in lacunarity values in the edge region between pretreatment and six months post-treatment ($P = 0.04$), with a decrease from 0.06 to 0.04. In the unaffected region, no significant difference was found in quantitative values between pretreatment and six months post-treatment ($P^{\text{fractal}} = 0.061$) ($P^{\text{lacunarity}} = 0.586$) ($P > 0.05$) (Table 2).

Analysis of fractal values revealed a significant difference between regions pretreatment ($P < 0.001$), with the center having a mean fractal value of 1.39, the edge 1.36, and the unaffected area 1.5 ().

**Table 2 Inter-regional comparison of quantitative values before and after cyst surgery.** The table shows the differences in fractal and lacunarity measurements between bone regions (center, edge, unaffected) in pre- and post-treatment phases.

| | Region | Pre-treatment Mean ± SD | Pre-treatment Median (Min–Max) | Test statistic/$p$ | Post-treatment Mean ± SD | Post-treatment Median (Min–Max) | Test statistic/$p$ |
|---|---|---|---|---|---|---|---|
| Fractal | Center | 1.39 ± 0,1[a] | 1.4 (1.21–1.58) | 10.351/**<0.001**[*] | 1.37 ± 0.1 | 1.36 (1.18–1.54) | 2.777/0.077[*] |
| | Edge | 1.36 ± 0.11[a] | 1.34 (1.19–1.56) | | 1.38 ± 0.15 | 1.37 (1.1–1.63) | |
| | Unaffected | 1.5 ± 0.1[b] | 1.5 (1.33–1.66) | | 1.45 ± 0.09 | 1.46 (1.27–1.6) | |
| Lacunarıty | Center | 0.06 ± 0.04 | 0.04 (0.02–0.17) | 1.529/0.465[**] | 0.06 ± 0.03 | 0.05 (0.02–0.14) | 4.353/0.113[**] |
| | Edge | 0.06 ± 0.03 | 0.06 (0.01–0.12) | | 0.04 ± 0.01 | 0.04 (0.02–0.08) | |
| | Unaffected | 0.05 ± 0.03 | 0.04 (0.02–0.15) | | 0.04 ± 0.02 | 0.04 (0.02–0.12) | |

**Notes.**
[*]Repeated measures analysis.
[**]Friedman test.
[a-b]There is no difference between groups marked with the same letter.
Lacunarity's interclass correlation coefficient (ICC) was determined to be 0.849 (95% CI [0.632–0.943]). Similarly, the fractal measurements' ICC of 0.716 (95% CI [0.373–0.887]) revealed a high degree of observer agreement. For both coefficients, the $p$-values were less than 0.001.
Bold values indicate statistically significant results ($P < 0.05$).

Post-treatment, there was no significant difference in mean fractal values among regions ($P = 0.077$). Lacunarity values did not show significant differences in median values among regions either pretreatment ($P = 0.465$) or post-treatment ($P = 0.113$) (Table 2).

## DISCUSSION

In this study, quantitative analysis of bone changes in patients undergoing cyst excision was performed using FD and lacunarity on digital panoramic radiographic images before and the 6th month after surgery. The 6th month follow-up interval was selected based on institutional protocol, aiming to allow sufficient time for observable bone remodeling while maintaining consistency across patient records. Before the treatment, FD values of unaffected area was the highest, followed by center and the edges of the cyst. Such difference was not detected between FD values of these regions at the 6th month post-treatment. Pre- and post-treatment lacunarity levels did not significantly differ between the regions. The edge region of the cyst showed a notable reduction in lacunarity, suggesting possible structural improvements, but the core region of the cyst and the unaffected bone regions did not show significant changes in FD and lacunarity values post-treatment.

In the study, FD and lacunarity values have been evaluated also on the symmetrical unaffected bone to validate our results. Indeed, the unaffected bone areas had the highest FD values prior to treatment, suggesting that the unaffected bone had more structural complexity than the cyst sections. The disparities in FD values between these regions, however, disappeared by the sixth month following surgery, indicating that the various regions' bone structures may have normalized or homogenized.

There were no discernible changes in lacunarity between the treated and untreated regions. Interestingly, there was a decrease in lacunarity at the cyst's edge region. The significant decrease in lacunarity at the lesion margin may reflect early bone remodeling activity, consistent with peripheral ossification patterns observed in healing bone (*Salhotra*

*et al., 2020*) This observation is consistent with clinical expectations of centripetal bone healing, where peripheral bone regeneration is typically more pronounced.

This could indicate improved bone healing and organization, possibly as a result of active remodeling taking place at the interface between the surrounding bone and the cystic lesion. It appears that the surgical intervention had a greater impact on the bone along the cyst's periphery than on the core region because the cyst's core did not show any appreciable changes in FD or lacunarity.

FA has proven to be a valuable tool in evaluating bone healing following maxillofacial surgical procedures. Research indicates that the utility of FA as a quantitative method for assessing osseous changes and monitoring healing processes in maxillofacial surgery. *Çolak et al. (2023)* has monitored the healing of the bone after orthognathic surgery, it is showed where FD typically decreases immediately post-operation and gradually increases during the healing process. This technique has been utilized to analyze trabecular changes in various regions, including mandibular osteotomy lines, condyles, and the angulus (*Çolak et al., 2023*; *Muftuoglu & Karasu, 2024*; *Heo et al., 2002*). Furthermore, FA has been applied to monitor osseous healing in mandibular defects treated with different materials, showing a strong correlation with histomorphometric data (*Nair et al., 2001*). *Kaba et al. (2022)* evaluated bone remodeling after alveolar crest augmentation with autogenous bone grafts prior to the implant placement. The study has succesfully demonstrate higher FD values of 5th month postoperatively compared to the 1st week following surgery both on the donor side and the graft recipient side.

FA allows for the precise assessment of bone healing and structural changes, providing more in-depth information compared to traditional radiographic evaluation methods. Astuti et al. reported a case of OKC treated with marsupialization, FD values acquired from panoramic radiograph has significantly increased on the periphery of the cyst area. *Lim et al. (2011)* conducted a study showing that bone healing increased significantly in the first few months after surgery and that the healing process continued up to the 12th month. *Koca et al. (2010)* evaulated preoperative and postoperative FD values of odontogenic cyst on panoramic radiographs of ten patients, found significant increase of FD values 18 months postoperatively. *Ozturk et al. (2022)* was evaluated the healing process of nonsyndromic odontogenic cysts after decompression treatment with FA. The measurements of FA from middle cyst and control area were done at the baseline, 1-month postoperative and the end treatment. They found an increase in FD values with time in central region of the cyst. After decompression, FD values were similar in the middle cyst and the control area, which inticated a successfull healing in the center of the cyst. Our findings largely confirm results from earlier FA study on bone healing, which also found region-specific differences in fractal parameters.

Recent studies have also applied fractal analysis to decompression and marsupialization cases, highlighting its broader applicability beyond enucleation. For instance, *Ozturk et al. (2022)* evaluated nonsyndromic odontogenic cysts treated with decompression and reported a progressive increase in fractal dimension values in the cyst center over time, indicating successful bone regeneration. Similarly, *Kaygisiz & Karsli (2024)* compared marsupialization and enucleation using FA and found comparable trabecular healing

patterns, especially in larger lesions treated conservatively. These findings support the versatility of FA in assessing bone healing across different surgical approaches.

In line with our findings, a recent study by *Tekin et al. (2025)* compared decompression and enucleation techniques in the treatment of odontogenic cysts using fractal analysis on panoramic radiographs. Their results demonstrated a significant increase in fractal dimension values at 3 and 6 months postoperatively in both treatment groups, with no statistically significant difference between them. This supports the notion that both surgical approaches promote comparable levels of trabecular bone regeneration. The study further highlights the reliability of fractal analysis as a quantitative tool for monitoring bone healing and reinforces the clinical relevance of our findings.

Another study estimated the FD values of DC, RC and OKC on CBCT images and found significant difference between the cyts, most high values were belonged to OKC. However, The study didn't evaluate the changes over time with treatment (*Murugesan, Vadivel & Ramalingam, 2024*). In our study we were unable to compare the FD values of different types of cysts because of the limited sample size.

These results highlight the practical value of FA as a quantitative, non-invasive way to track bone repair following cyst surgery. Predicting long-term results and customizing postoperative therapy may be made easier with the ability to identify minute changes in bone microarchitecture. The notable decrease in lacunarity at the edge region implies that FA may be especially helpful in identifying regions where bone remodeling is currently occurring.

The application of consistent imaging methods and meticulous statistical analyses are two of this study's strong points.

It should be the goal of future studies to confirm these results with larger cohorts and longer follow-up times. Furthermore, investigating how to combine FA with additional imaging modalities or biomarkers may offer a more thorough comprehension of the dynamics of bone repair.

Future studies should consider incorporating advanced imaging modalities such as cone beam computed tomography (CBCT), densitometry, or scintigraphy to provide more comprehensive three-dimensional and metabolic assessments of bone healing.

Recent advances in radiographic texture analysis have broadened the scope of bone healing assessment beyond traditional fractal analysis. For instance, *Lopes et al. (2025)* utilized radiomics-driven CBCT texture analysis to quantify periapical bone healing, demonstrating its potential as a biosensor for tissue regeneration. Similarly, *Bayat et al. (2024)* applied texture analysis to evaluate hard tissue changes following socket preservation procedures, revealing significant differences in bone quality based on graft material. Moreover, *Park et al. (2024)* highlighted the diagnostic value of CBCT-based texture analysis in differentiating between cemento-osseous dysplasia and periapical cysts. These studies underscore the growing utility of advanced radiographic texture analysis in both diagnostic and healing-monitoring contexts within maxillofacial surgery.

### Limitations

This study has several limitations, including its retrospective design, small sample size, lack of control for clinical confounders such as smoking and potential variability in ROI selection. Future prospective studies with larger cohorts and standardized protocols are needed to validate these findings. The small sample size limited our ability to perform subgroup analysis based on cyst type, which restricts the generalizability of specific findings. Additionally, due to the retrospective nature of the study, no a priori power analysis was performed, which was considered a limitation.

## CONCLUSIONS

In conclusion fractal analysis has shown to be an effective method for assessing bone repair following odontogenic cyst surgery. The reduction in lacunarity values found in the edge region underscores the possibility that this technique can provide comprehensive insights into bone, regeneration processes, ultimately leading to better patient care and clinical results.

Although currently used for research purposes, fractal analysis holds promise as a future clinical tool for non-invasive monitoring of bone healing and treatment planning.

### Funding

The authors received no funding for this work.

### Competing Interests

The authors declare there are no competing interests.

### Author Contributions

- Ayse Tas conceived and designed the experiments, performed the experiments, analyzed the data, prepared figures and/or tables, authored or reviewed drafts of the article, and approved the final draft.
- Elif Celebi conceived and designed the experiments, performed the experiments, analyzed the data, prepared figures and/or tables, and approved the final draft.
- Zeynep Çukurova Yilmaz conceived and designed the experiments, performed the experiments, authored or reviewed drafts of the article, and approved the final draft.

### Human Ethics

The following information was supplied relating to ethical approvals (i.e., approving body and any reference numbers):

ISTANBUL MEDIPOL UNIVERSITY

Presidency of Non-Interventional Clinical Research Ethics Committee.

Faculty of Medicine Clinical Research Ethics Committee (Approval No: E-10840098-202.3.02-2244, 2024).

## Data Availability

The raw data is available in the Supplementary File.

## Supplemental Information

Supplemental information for this article can be found online at http://dx.doi.org/10.7717/peerj.19745#supplemental-information.

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
