# Peer review of "Assesment of bone healing after surgical management of odontogenic cysts utilizing fractal analysis—a retrospective cross-sectional study"

_PeerJ, doi:10.7717/peerj.19745_

## Round 0.1 · original submission · Major Revisions

The manuscript must be subject to revisions according to the reviewers' criticisms.

**Language Note:** The review process has identified that the English language must be improved. PeerJ can provide language editing services - please contact us at [email protected] for pricing (be sure to provide your manuscript number and title). Alternatively, you should make your own arrangements to improve the language quality and provide details in your response letter. – PeerJ Staff

·

Basic reporting

First of all, I want to mention that this is a very interesting study, and I congratulate you for taking the time to investigate the behavior of this type of lesion. Secondly, I have some questions:

Experimental design

You mention the evaluation of 17 patients. The number seems a bit small to draw conclusions, especially considering that within the 17 patients there are three different types of cysts that do not behave the same. It would be ideal to continue this work to include a larger number of patients and obtain more reliable conclusions to inform the treatment of each type of cyst.

Considering the difficult behavior and the justification for requesting follow-up cone beam CT, what is the reason for not performing measurements at 6 months with a 3D image? Panoramic radiography allows us to observe changes in the extension of the lesion in the anteroposterior and vertical directions, but not in depth. Therefore, I think it is appropriate to repeat this study with cone beam CT.

I'm not sure if the cysts were previously biopsied for diagnosis. Regarding treatment, were they decompressed or was a surface treatment performed? Was the treatment mechanical or chemical? Could you please answer these questions? These answers can greatly explain the results obtained.

While I think it's a great contribution to 2D imaging, I believe objective measurement with a cone beam is essential in the three dimensions of space. Is it feasible to perform the same procedure with a cone beam? Densitometry? Scintigraphy?

Validity of the findings

Has this type of measurement guided your decision-making?

It would be interesting to obtain a biopsy in cases requiring implant rehabilitation.

What was the reason for evaluating changes at 6 months instead of 3 months? Considering that bone changes can be observed on X-rays as early as 3 months.

Additional comments

In my experience, peripheral bone regeneration is expected, since bone healing behaves centripetally.

Reviewer 2 ·

Basic reporting

The manuscript addresses a current issue in maxillofacial surgery and radiology. The methodology of the study is suitable for publication.

However, the Discussion section may benefit from current literature. In particular, the authors' discussion of the findings of Tekin et al. (2025), who compared decompression and enucleation using fractal analysis in odontogenic cysts, will increase the consistency and awareness of the article.

Tekin G, Kosar YC, Saruhan Kose N, Dereci O, Ozkiris S, Incebeyaz B. Comparison of Decompression and Enucleation Using Panoramic Radiography and Fractal Analysis for Odontogenic Cysts. Med Sci Monit. 2025 Mar 3;31:e947910. doi: 10.12659/MSM.947910.

Additionally, shape quality and annotation (e.g., ROI representation in Figure 1) can be improved for better clarity and reproducibility.

Experimental design

The inclusion and exclusion criteria of the study were not written. G Power analysis was not performed. These need to be re-evaluated.

Validity of the findings

Statistical analysis was appropriate and both parametric and nonparametric methods were used depending on the distribution of the data. However, although the decrease in lacunarity at the margin was statistically significant, the clinical significance of this finding has not been adequately explained. It would be useful to relate this to expected bone remodeling patterns, perhaps using other studies on healing kinetics. Additionally, the inability to perform subgroup analysis based on cyst type due to the small sample size limits the depth of interpretation. This limitation should be discussed explicitly in the results or limitations section.

Additional comments

Authors should expand the discussion to include comparative findings from recent fractal analysis studies beyond just surgical excision, such as decompression or marsupialization methods.

The manuscript requires grammar editing for better flow and clarity (e.g., “lac values” should be written consistently as “lacunarity values”).

If possible, including a flowchart of the patient selection process may increase transparency of the study design.

Reviewer 3 ·

Basic reporting

This paper investigated bone healing following surgical removal of odontogenic cysts by applying FD analysis to panoramic radiographs. The study aims to correlate fractal values with cyst size and healing outcomes, offering a quantitative approach to bone regeneration assessment.
However, the manuscript requires major revisions before it can be considered for publication.

Materials and Methods:
Were all panoramic radiographs acquired using the same machine and exposure settings? Given the sensitivity of fractal analysis to image quality and resolution, please clarify how acquisition variability was controlled.
How were ROIs standardized across patients and timepoints? Were they selected manually or with software assistance? Did the authors assess interobserver or intraobserver variability?
What was the postoperative interval between cyst enucleation and the follow-up radiograph used for analysis? Were healing timepoints standardized or did they vary significantly?
Were patient-related factors such as age, smoking status, systemic disease, or use of bone grafts considered or controlled for? These may influence bone healing outcomes.

Discussion:
The discussion section would benefit from deeper contextualization within recent literature using radiographic texture analysis to monitor bone healing. In particular, the authors should consider citing the following study:

Lopes et al. Radiomics-Driven CBCT Texture Analysis as a Novel Biosensor for Quantifying Periapical Bone Healing: A Comparative Study of Intracanal Medications. Biosensors (Basel). 2025.9;15(2):98.
This reference is relevant because it applies advanced texture analysis techniques (radiomics) to CBCT for assessing bone healing after endodontic procedures. Although the imaging modality and clinical context differ, the conceptual framework—using imaging biomarkers to quantify healing—is closely aligned with the current study.

The authors should include a dedicated limitations section to enhance transparency and scientific rigor.

Experimental design

Were all panoramic radiographs acquired using the same machine and exposure settings? Given the sensitivity of fractal analysis to image quality and resolution, please clarify how acquisition variability was controlled.
How were ROIs standardized across patients and timepoints? Were they selected manually or with software assistance? Did the authors assess interobserver or intraobserver variability?
What was the postoperative interval between cyst enucleation and the follow-up radiograph used for analysis? Were healing timepoints standardized or did they vary significantly?
Were patient-related factors such as age, smoking status, systemic disease, or use of bone grafts considered or controlled for? These may influence bone healing outcomes.

Validity of the findings

The findings offer an interesting insight into the use of fractal dimension analysis for assessing bone healing after cyst enucleation. However, the study has methodological constraints that affect the validity of the results—particularly the retrospective design, lack of control for clinical confounders, absence of intra-/interobserver agreement analysis, and potential variability in radiographic acquisition protocols.

---

## Round 0.2 · accepted · Accept

The authors have addressed the reviewers' comments. The manuscript is ready for publication.

Reviewer 2 ·

Basic reporting

The author has made the requested revisions.

Experimental design

The author has made the requested revisions.

Validity of the findings

The author has made the requested revisions.

Additional comments

The author has made the requested changes. It can be published.

Reviewer 3 ·

Basic reporting

The authors have satisfactorily addressed my concerns.

Experimental design

no comment

Validity of the findings

no comment